# Measuring Self-Efficacy for Exercise among Older Adults: Psychometric Properties and Measurement Invariance of a Brief Version of the Self-Efficacy for Exercise (SEE) Scale

**DOI:** 10.3390/healthcare12161642

**Published:** 2024-08-17

**Authors:** James Dawe, Elisa Cavicchiolo, Tommaso Palombi, Roberto Baiocco, Chiara Antoniucci, Jessica Pistella, Guido Alessandri, Lorenzo Filosa, Simone Tavolucci, Anna M. Borghi, Chiara Fini, Andrea Chirico, Fabio Alivernini, Fabio Lucidi

**Affiliations:** 1Department of Developmental and Social Psychology, Sapienza University of Rome, 00185 Rome, Italy; tommaso.palombi@uniroma1.it (T.P.); roberto.baiocco@uniroma1.it (R.B.); chiara.antoniucci@uniroma1.it (C.A.); jessica.pistella@uniroma1.it (J.P.); andrea.chirico@uniroma1.it (A.C.); fabio.alivernini@uniroma1.it (F.A.); fabio.lucidi@uniroma1.it (F.L.); 2Department of Systems Medicine, University of Rome Tor Vergata, 00133 Rome, Italy; elisa.cavicchiolo@uniroma2.it; 3Department of Psychology, Sapienza University of Rome, 00185 Rome, Italy; guido.alessandri@uniroma1.it (G.A.); lorenzo.filosa@uniroma1.it (L.F.); simone.tavolucci@uniroma1.it (S.T.); 4Department of Dynamic and Clinical Psychology, and Health Studies, Sapienza University of Rome, 00185 Rome, Italy; anna.borghi@uniroma1.it (A.M.B.); chiara.fini@uniroma1.it (C.F.)

**Keywords:** self-efficacy, self-efficacy for exercise, SEE, older adults, cross-cultural, validation

## Abstract

(1) Background: Physical activity is known to promote health and psychological well-being in older adults, yet global inactivity rates in this population remain high. Among the factors associated with physical activity, self-efficacy for exercise represents a key predictor for developing effective interventions in older adults. This study aimed to validate the Self-Efficacy for Exercise Scale (SEE) in individuals over 65. (2) Methods: A sample of 726 older adults from the USA and Italy (51.1% females; age range = 65–95 years; Mage = 72.57, SDage = 6.49) completed the SEE, along with the Godin–Shepard Leisure-Time Physical Activity Questionnaire (GSLTPAQ), the Big Five Inventory 2—Extra Short Form (BFI-2-XS), and the 12-item Short Form Health Survey (SF-12). (3) Results: The SEE showed a Cronbach’s Alpha of 0.88 and a Composite Reliability of 0.89. Moreover, it demonstrated a unidimensional factor structure and good fit indices. Full measurement invariance was achieved across gender and age, while partial scalar invariance was found across countries, suggesting minor cultural differences. Correlation with the GSLTPAQ, the BFI-2-XS, and the SF-12 support the convergent and nomological validity of the SEE. (4) Conclusions: These findings provide evidence that the SEE is a reliable and valid measure of self-efficacy for exercise among older adults and that the items are interpreted similarly across different ages, genders, and cultures.

## 1. Introduction

Past studies have demonstrated that physical activity can significantly improve both physical and psychological health in older adults. For example, it can reduce the risk of chronic diseases such as diabetes and heart disease [1,2], lower the likelihood of depression [3,4,5], and decrease the risk of neurodegenerative diseases like Alzheimer’s [6] and dementia [3]. Despite the well-known benefits of engaging in physical activity, the World Health Organization (WHO, [7]) expressed significant concerns regarding the high global levels of physical inactivity and its impact on economies. For instance, it is estimated that 47% of elder males in the U.S. and 52% in Italy, along with 65% of elder females in Italy and 62% in the U.S., are physically inactive [7]. These percentages are notably higher compared to younger adults, emphasizing the urgency of increasing exercise adherence among older adults—the population that stands to gain the most from physical activity. Therefore, enhancing engagement in physical activities within this population is of primary importance.

To further understand exercise behavior among older adults, past studies have identified several factors associated with physical inactivity. For instance, Sullivan and Lachman [8] highlighted three types of barriers: demographic (e.g., low SES, being an older male), environmental (e.g., inclement weather, unpleasant scenery), and psychosocial (e.g., embarrassment, fear of failure, low self-efficacy). Similarly, Baert et al. [9] categorized barriers to physical activity into three domains: intrapersonal (e.g., health/physical impairment, low expectations), interpersonal (e.g., lack of social support), and community (e.g., cost, lack of transportation). Among these factors, motivational variables such as self-efficacy for exercise have been identified as one of the most important predictors of exercise behavior [10,11,12,13,14,15]. Therefore, identifying and assessing self-efficacy for exercise is an important step in the development of interventions aimed at increasing exercise participation among older populations.

### 1.1. The Concept of Self-Efficacy in Physical Activity

In the context of physical activity, self-efficacy for exercise refers to the belief in one’s ability to exercise safely and successfully [15,16,17]. This belief is a critical factor in behavior change and plays a decisive role in starting and maintaining adherence to health behaviors and exercise programs over the long term [8,16,18,19,20]. Indeed, elders with low self-efficacy for exercise have been described as hesitant to engage in new exercises, likely to abandon them, quick to surrender when challenges arise, and prone to focus on their limitations [21]. Conversely, those with high self-efficacy for exercise are characterized by their ability to set and pursue personal goals, recover from setbacks, and confidently engage in exercises [21]. Therefore, developing strategies to enhance self-efficacy for exercise may be crucial in promoting physical activity and well-being among older adults, a population that stands to gain significantly from such interventions.

### 1.2. Brief Version of the Self-Efficacy for Exercise Scale (SEE)

Resnick and Spellbring [22] have proposed a nine-item questionnaire for the assessment of self-efficacy for exercise, developed specifically for the older adult population: the Self-Efficacy for Exercise Scale (SEE). This scale measures self-efficacy levels in engaging in physical activity despite environmental, physical, and psychological barriers such as body pain and feelings of depression. The U.S. version of the test demonstrated good reliability with an internal consistency of 0.92, a well-fitting factor structure, and successfully predicted exercise levels [22,23]. However, whether these psychometric properties hold in other cultures remains to be verified. For example, Lee et al. [17] identified differences between the Chinese and U.S. versions of the SEE. Indeed, the Chinese version showed lower model fits (though still acceptable), reduced internal consistency (0.92 in the U.S. vs. 0.75 in China), and lower item reliability. Similar results were observed among U.S. minority elders by Resnick et al. [13]. These discrepancies align with general literature studies suggesting that cultural influences on questionnaire characteristics are quite common [24,25,26]. To our knowledge, the psychometric properties of the cross-cultural version of the SEE are still unknown. Given the cultural differences in the psychometric proprieties of the SEE highlighted in past studies [13,17], before the questionnaire can be applied more broadly, a cross-cultural evaluation is recommended.

To further generalize the use of the questionnaire in different contexts and situations, we analyzed a brief version of the SEE.

### 1.3. SEE Measurement Invariance and Cultural Differences

To ensure that a tool measures the same constructs consistently across groups with different individual characteristics (e.g., age, gender) and cultures, measurement invariance needs to be tested. This is particularly true when considering cultural factors, where differences between the psychometric proprieties of a given questionnaire across countries are often the norm rather than the exception [24,25,26]. For example, North American elders tend to give great value to personal achievement [27] and physical activity [28,29], while among European elders, those values seem to be less relevant [30,31,32]. These differences could lead North American elders to underestimate the relevance of some barriers compared to their European counterparts, impacting the importance given to some of the items regarding self-efficacy for exercise. For example, to meet cultural values, North American elders could be more inclined to engage in physical activity, which may lead them to give less importance to items related to the barrier of feeling tired. Moreover, age and gender differences could also impact the importance given to some of the items. For example, past research has highlighted that loss of energy and fatigue increase with age [33,34,35]. Therefore, for young-old adults (65–75 years old) the barrier of feeling tired may be perceived as less important than for old-old adults (over 75). Finally, there is substantial evidence of gender differences concerning engagement in physical activity and perceived barriers. For example, women prefer physical activity that involves social interactions [36], engage more in household activities, perceive their health as poorer, and encounter more barriers than men [37]. These differences could be reflected in how self-efficacy for exercise is conceptualized and how SEE items are interpreted across gender and age.

To demonstrate that elders with different characteristics or from diverse cultures perceive self-efficacy for exercise as a unidimensional construct and interpret its indicators consistently, establishing measurement invariance is essential. This process typically involves three steps [26]. Firstly, it must be established that elders with different characteristics conceptualize self-efficacy for exercise as a unidimensional construct (configural invariance), with all items loading on a single latent factor. Once this level of invariance is established, the next step is to test metric invariance, which ensures that all items are interpreted similarly across elders with different characteristics. The final step (scalar invariance) is to ensure that regardless of group membership, elders with the same level of self-efficacy for exercise respond similarly to the corresponding items.

Problems at any step of this process have consequences on the statistical analyses that can be performed. For instance, both configural and metric invariance are necessary to conduct correlational or cross-lagged panel model analyses [24,38]. If these levels of invariance are not met, no further analyses can be conducted [38], as the questionnaire would then be measuring different constructs in different ways across groups. Conversely, if meaningful group comparisons are the goal, scalar invariance is required [39]. However, scalar noninvariance is frequently observed [25,26,40], leading to the common practice of testing for partial scalar invariance by freeing some item intercepts to vary across groups. While not the ideal solution, this approach enables group comparisons and provides valuable insights into group differences [41,42,43].

The USA and Italy represent two countries with distinct cultures, social structures, and retirement plans, which may reflect differences in how barriers to physical activity are experienced. For example, the USA is recognized as a highly individualistic culture [44], where personal achievements, physical activity, and self-fulfillment are valued. In contrast, Italian culture is considered less individualistic and more collectivistic than that of the USA [44].

### 1.4. The Present Study

Considering the significant role self-efficacy for exercise plays in motivating elders to engage in physical activity and healthy behaviors [11,12,13,14,15], the general aim of this study was to validate and establish the psychometric properties of a brief version of the Self-Efficacy for Exercise Scale (SEE) with a sample of older adults for the first time. Indeed, past studies have highlighted that barriers to physical activity are not equivalent for all individuals and can vary according to geographical regions, gender, preferences, and attitudes [36,38,45,46,47]. These differences, if not considered, can lead to an over- or underestimation of the levels of Self-Efficacy for Exercise in some groups. For example, the relevance of SEE item 1, which is related to being able to exercise when the weather is bothersome, strongly depends on the geographical region [45]. Indeed, the impact of frequent bad weather on self-efficacy for exercise differs significantly from regions where it is rare. In addition, this item does not cover indoor activities, which are preferred by older women [38]. Item 2, which refers to feeling bored by the activity program, does not represent a barrier for individuals who prefer to engage in unstructured exercises (e.g., walking, gardening), such as older adults [46], where there is generally no program to follow. Item 4 refers to exercising alone, which is influenced by gender differences in exercise behaviors among older adults. Indeed, older men prefer exercising alone, whereas older women prefer social activities and have greater social motives for being active [36,47]. For the aforementioned reasons, as well as to extend the generalizability of the scale to different groups, the three items from the original version of the SEE were not included. Finally, items 8 (feeling stressed) and 9 (feeling depressed) do not cover different experiences; therefore, it is more comprehensive if combined into a single item concerning feeling in a bad mood. The final version of the questionnaire encompassed five items.

More specifically, the first aim of the present study was to explore the factor structure underlying the SEE, hypothesizing a unidimensional structure, as posited by the literature [22].

Our second aim was to test measurement invariance across cultural contexts (i.e., U.S. vs. Italy), genders, and age groups (i.e., 65–75 years old vs. over 75). In accordance with previous studies that have highlighted changes in physiological and psychological patterns between the two groups, we divided older adults into young-old adults (65–75) and old-old adults (over 75) [48,49,50,51,52,53]. We hypothesize, based on the existing literature, that scalar measurement invariance will be established for gender and age groups [25]. However, due to the potential influence of cultural factors [25], some differences might emerge between the U.S. and Italy.

The third aim was to analyze the convergent validity of the SEE through the Average Variance Extracted (AVE) and the correlation with the Godin–Shepard Leisure-Time Physical Activity Questionnaire (GSLTPAQ) [54], a measure of physical activity. In line with the previous literature (e.g., [11,13]), we expected a small positive correlation between self-efficacy for exercise and levels of physical activity.

Past studies have theorized that self-efficacy for exercise is associated with mental and physical health, as well as with personality traits [17,22,55,56]. Indeed, some of these variables can represent barriers to physical activity (e.g., neuroticism or poor health), leading to lower levels of self-efficacy for exercise, while others can facilitate engagement in physical activity (e.g., extraversion), thereby enhancing self-efficacy for exercise. Therefore, the fourth aim of this study was to analyze the nomological validity of the SEE in relation to these variables, measured through the 12-item Short Form Health Survey (12-SF) [57] and the Big Five Inventory-2 Extra Short Form (BFI-2-XS) [58]. According to past studies [17,22,55,56], we hypothesize a small positive correlation of self-efficacy for exercise with mental health, physical health, conscientiousness, and extraversion and a moderate negative correlation with negative emotionality.

## 2. Materials and Methods

### 2.1. Sample and Procedure

The present study employs a cross-sectional design, and the data were collected from 726 older adults (over 65), 51.1% of whom were females. The sample was composed of 343 elders living in the United States of America (U.S.) and 383 living in Italy (ITA). The average age was 72.57 years (SD = 6.49; min = 65, max = 95), with 60.5% being 65–75 years old and 39.5% being above 75 years old. Concerning the individual countries, 48.7% of the U.S. sample and 51.4% of the Italian sample were females. The average age of the U.S. older adults was 73.21 years (SD = 7.23; min = 65, max = 94), while the average age of the Italian older adults was 71.86 years (SD = 5.46; min = 65, max = 95). No gender differences were observed across countries (χ^2^ = 0.547, df = 1, *p* = 0.460), while the Italian sample showed a higher mean age than the U.S. sample (F = 8.07, *p* > 0.01).

The study was coordinated by a group of Italian researchers and consisted of a questionnaire on health, personality traits, and Self-Efficacy for Exercise in older people. Recruitment and data collection were conducted through the Prolific website and took place between April and September 2023. All participants voluntarily signed up for the study and filled out an online survey, which took an average of 12 min to complete. The exclusion criteria were as follows: current or past neurological disorder or major medical illness (e.g., dementia, traumatic brain injury, schizophrenia, epilepsy, active nausea, vomiting), current psychiatric disorder (e.g., major depression), or a severe sensory or motor deficit that would preclude physical activity or exercise.

### 2.2. Measures

The Self-Efficacy for Exercise Scale (SEE) [22] is a self-report questionnaire composed of nine items for the assessment of self-efficacy for exercise (e.g., how confident the individual feels about exercising when feeling tired), specifically designed for older adults. Each item is rated on an 11-point scale (from 0 = “Not confident” to 10 = “Very confident”). The scale reflects the degree of confidence that an elder has in engaging in exercise in the presence of specific barriers (e.g., a bad mood). We tested a shorter version of the questionnaire to maximize the generalizability of the questionnaire. Some of the items were excluded because they could lead to an underestimation or overestimation of the self-efficacy for exercise in some contexts or groups. The final Italian version of the SEE was therefore composed of five items. Special care was taken in the process of adapting the items of the SEE from English to Italian. First, they were translated from English into Italian by the authors and then back-translated by a person fluent in both languages. Second, a team of independent judges reviewed the original and back-translated versions of the scale until an agreement on their consistency was reached.

The Godin–Shephard Leisure-Time Physical Activity Questionnaire (GSLTPAQ) [54] is a widely used self-report questionnaire composed of three items for assessing leisure-time physical activity, i.e., any activity performed during one’s free time that increases energy expenditure [59]. Each item represents a group of different types of exercise: strenuous (e.g., running), moderate (e.g., fast walking), and mild (e.g., yoga). The participant indicates how many times per week they engage, during free time, in 20 min of strenuous, moderate, and mild exercise, respectively. To compute the leisure score index (LSI), the time per week for each item is multiplied by the Metabolic Equivalent Task (MET) value required by the corresponding exercise (i.e., 3 for mild, 5 for moderate, and 9 for strenuous). Previous studies have provided support for the GSLTPAQ’s validity through correlations with physical health measures such as VO2max, body fat percentage, and fitness center attendance records [60]. Test-retest reliability showed a coefficient of 0.65 [60].

The Big Five Inventory-2 Extra Short (BFI-2-XS) [58] is composed of 15 items rated on a 5-point Likert scale (from 1 = “Disagree strongly” to 5 = “Agree strongly”). It consists of 5 scales and 15 subscales: extraversion (sociability, assertiveness, and energy level), agreeableness (compassion, respectfulness, and trust), conscientiousness (organization, productiveness, and responsibility), negative emotionality (anxiety, depression, and emotional volatility), and open-mindedness (aesthetic sensitivity, intellectual curiosity, and creative imagination). However, due to its brevity, it is not recommended for assessing the finer-grained facet traits within each domain [58]. Previous research supported the five-factor structure, providing evidence for external validity, alpha reliability (from 0.73 for negative emotionality to 0.55 for agreeableness), and a part–whole correlation ranging from 0.92 for extraversion and negative emotionality to 0.87 for agreeableness [58].

The 12-item Short Form Health Survey (SF-12) [57,61] is a widely used questionnaire aimed at assessing health-related quality of life. It has been translated into many languages and is practical to use. It includes 12 questions that reflect various aspects of physical health, such as physical functioning, bodily pain, and general health perceptions, and mental health, such as vitality, social functioning, and emotional role limitations. Each item is used to compute the Physical Component Summary (PCS) score and the Mental Component Summary (MCS) score. Ware et al. [57] provided the algorithms for computing the raw scores and for standardizing PCS and MCS into T scores, which are used in the present study. Past research involving older adults has shown good reliability (range 0.73–0.86) and acceptable convergent and discriminant validity [62,63].

### 2.3. Data Analysis

Based on the total sample of older adults, a confirmatory factor analysis (CFA) was conducted to analyze the factor structure of the SEE (Figure 1), testing a model with all items loading on one factor. Model goodness-of-fit was assessed using the following indices: Satorra–Bentler Chi-squared test, root mean square error of approximation (RMSEA), Comparative Fit Index (CFI), Tucker–Lewis Index (TLI), and standardized root mean square residual (SRMR). Widely accepted rules of thumb for good and acceptable fits were applied to evaluate the model’s fit to the data [64,65]. Once the factor structure was established, measurement invariance (Configural, Metric, and Scalar) across elders with different characteristics (gender and age) living in either the U.S. or Italy was tested through a series of multigroup confirmatory factor analyses (MGCFAs). This involved sequential steps testing increasingly restrictive models by imposing equality constraints on patterns of factor loadings, item factor loadings, and item intercepts [24]. The more restricted model was compared with the less restricted one. Loss of fit was tested using the ΔSatorra–Bentler Chi-squared test, ΔCFI, and ΔRMSEA [66], with changes greater than 0.01 and 0.015, respectively, indicating noninvariance [66]. Where noninvariance was found, we adjusted models based on modification indices (MIs) and theoretical considerations. All analyses utilized Maximum Likelihood estimation with robust standard errors (MLR) for parameter estimation.

Internal consistency reliability was analyzed through Cronbach’s Alpha and the Composite Reliability (CR). Both use the same rule of thumb: values between 0.60 and 0.70 are acceptable, values between 0.70 and 0.90 are satisfactory, and values above 0.90 are indicative of redundancy of the items, and are therefore not desirable [67,68].

Finally, the Average Variance Extracted (AVE) and Pearson’s correlations between the SEE and the Big Five factors and mental health (MCS) and physical health (PCS) were computed, alongside a Spearman’s correlation with the leisure score index (LSI) (due to violation of normality assumption) to provide evidence of convergent and nomological validity. The use of the factor scores of the scale, instead of the sum or mean of the observed indicators, helps to reduce the impact of measurement error on the correlations.

MPlus 8 [69] was used to conduct the CFA and MGCFA, while Pearson correlation and Cronbach’s Alpha were computed with SPSS 27. Finally, the AVE and the Composite Reliability (CR) were computed using an Excel spreadsheet (Online Resource 2).

## 3. Results

Descriptive statistics of the SEE items, which are reported in Table 1, showed values for skewness and kurtosis within the tolerable range of ±2, supporting the assumption of normality [70]. Descriptive statistics for the GSLTPAQ, BFI-2-XS scales, and the two measures of mental health (MCS12) and physical health (PCS12), as well as their corresponding items, are reported in Appendix A. Given the deviation of leisure score index (LSI) from normality (kurtosis = 5.568), we decided to use Spearman’s rho correlations.

All fit indices, except for the chi-squared (probably due to its sensitivity to sample size), indicated that the proposed unidimensional model for the SEE fits the empirical data well: χ^2^(5) = 21.833, *p* < 0.001; RMSEA = 0.068; CFI = 0.984; TLI = 0.967; SRMR = 0.019. Figure 1 shows that the standardized factor loadings ranged from 0.67 to 0.85. Residuals ranged from a minimum of 0.271 for item 4 to a maximum of 0.547 for item 1.

**Figure 1 healthcare-12-01642-f001:**
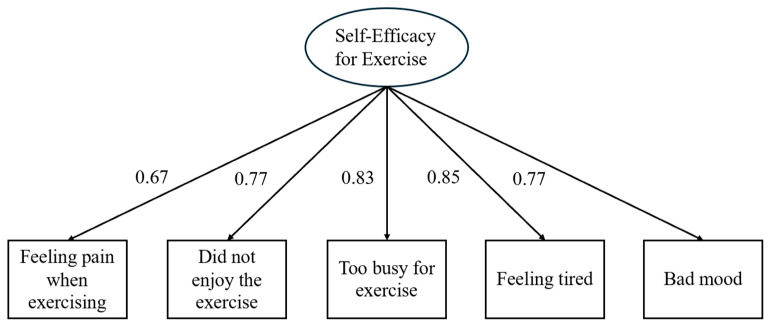
Confirmatory factor analysis results. Note: all the values are standardized, and the measurement errors, which were not displayed, are not allowed to correlate.

### 3.1. Measurement Invariance

The results of the multigroup confirmatory factor analysis (MGCFA) across elders with different characteristics (i.e., gender, age group) or living either in the U.S. or Italy are reported in Table 2. When the fit indices of the more restricted model (i.e., Metric) were compared with the less restricted model (i.e., Configural), the ΔCFI and ΔRMSEA showed minor loss of fit, supporting metric invariance (for country, ΔRMSEA = 0.003, ΔCFI = −0.010; for gender, ΔRMSEA = −0.004, ΔCFI = −0.005; for age, ΔRMSEA = −0.004, ΔCFI = −0.004). This result indicates that the model reproduced the data well, despite the pattern of factor loadings and the items’ factor loadings being restricted in order to be equal across groups. It must be noted that to archive good fit indices for the configural model when age groups were compared, item 2 and item 5 in the over 75 group were free to covary.

Concerning full scalar invariance, when comparing metric invariance models with a model in which all intercepts are restricted to be invariant across groups, ΔCFI and ΔRMSEA showed minor loss of fit only for gender (ΔRMSEA = 0.000, ΔCFI = −0.006) and age (ΔRMSEA = −0.004, ΔCFI = −0.003). The results do support partial scalar invariance when we compare U.S. and Italian older adults, highlighting some cultural influences. To achieve a model with acceptable fit indices, the intercepts of item 3 (Too busy for exercise) and item 4 (Feeling tired) were left free to vary. Given that the majority of the items’ intercepts are invariant, the standards for partial scalar invariance were met [71,72].

### 3.2. Reliability and Convergent Validity

Concerning the reliability of the SEE, Cronbach’s Alpha (0.88) and the CR (0.89) are both higher than the threshold of 0.70, suggesting that the SEE is a reliable measure of self-efficacy for exercise.

Concerning convergent validity, the AVE for the SEE is 0.61, which exceeds the threshold of 0.50 [67]. This result implies that this factor explains 61% of the variance in its associated items. Moreover, the SEE has shown a large positive correlation with the leisure score index (LSI) (0.531, *p* < 0.001). These results support the convergent validity of the SEE.

### 3.3. Nomological Validity

The results show a positive correlation between the SEE and physical health with a moderate effect size (PCS; 0.353, *p* < 0.001), as well as a positive correlation with mental health with a low effect size (MCS; 0.209, *p* < 0.001). These results suggest that a higher level of confidence in one’s ability to exercise safely and successfully is associated with the perception of having good physical and mental health. Concerning the relationship with the BFI-2-XS scales, the SEE shows a positive correlation, all with small effect size, with extraversion (0.082, *p* < 0.05), agreeableness (0.097, *p* < 0.01), conscientiousness (0.130, *p* < 0.001), and open-mindedness (0.253, *p* < 0.001), as well as a negative correlation with negative emotionality (−0.186, *p* < 0.001). These associations suggest that older adults with a high degree of SEE tend to be more sociable, full of energy, warm, organized, dependable, creative, open to new ideas and experiences, and emotionally stable.

## 4. Discussion

Self-efficacy for exercise represents an important factor among older adults, influencing their health and exercise behavior [11,15,16,18,73]. Due to its relevance, Resnick and Spellbring [22] developed the Self-Efficacy for Exercise (SEE) scale specifically for older adults. However, the psychometric properties of the scale in the Italian population are so far unknown. Indeed, whether these psychometric properties hold in other cultures needs to be verified before the questionnaire can be applied more broadly. Based on a cross-cultural sample of U.S. and Italian older adults (over 65), the present study aimed to investigate the reliability and validity of a brief version of the SEE, which offers a distinct advantage in terms of its feasibility.

### 4.1. Factor Structure and Measurement Invariance

First, we examined the factor structure of the SEE. Our findings show that the hypothesized structure of the brief version of the SEE had a good fit to the data, with all items loading on one factor. This finding aligns with previous studies on the full version of the SEE, which have highlighted the unidimensional structure of the questionnaire [73,74].

Second, for the first time, we assessed the measurement invariance of the SEE across older adults with different characteristics (i.e., gender and age group) and cultural backgrounds (U.S. vs. Italy). This is a fundamental part of the validation process, given the possible impact of group differences on the conceptualization of self-efficacy for exercise and in the interpretation of its items [10,26,49]. The results of the present study support the full scalar invariance of the SEE across gender and age (though item 2 (“did not enjoy the exercise”) and item 5 (“bad mood”) were free to covary in the over 75 groups), allowing for comparisons between these groups on SEE scores. Although the past literature has reported that distinct groups of older adults (e.g., older males vs older females) experience barriers to physical activity differently [75,76,77], our results suggest that regardless of their characteristics (i.e., gender and age group), they interpreted the items and conceptualized self-efficacy for exercise consistently.

Concerning measurement invariance across older adults of different nationalities (U.S. vs. Italy), only partial scalar invariance was achieved. This enables the comparison of the strength of relationships (e.g., covariances, correlations, regression) between latent variables across groups [24,25] and the study of mean group differences, which requires scalar invariance or at least partial scalar invariance [26,41,42,43]. Therefore, older adults living in different countries conceptualize and interpret the items similarly; however, similar responses from U.S. and Italian individuals to items 3 and 4 are not indicative of the same levels of self-efficacy for exercise. These results suggest the presence of minor cultural differences. Indeed, in the present study, item 3 (Too busy for exercise) and item 4 (Feeling tired) of the SEE showed variation across the two groups, with U.S. older adults scoring higher on average than Italians on feeling able to exercise when they are too busy or they feel tired. The cultural tendency of U.S. elders to represent themselves as physically active, younger, and full of energy [29,30,32] could explain the higher level of confidence in being able to exercise even when they face the barrier of being too busy or feeling tired, compared to their Italian counterparts. This finding aligns with the previous literature, in which older adults from different cultures reported distinct barriers to physical activity [4,78,79].

### 4.2. Convergent and Nomological Validity

Our third aim was to assess the convergent validity of the SEE. The results reveal that most of the variance (61%) in the indicators of self-efficacy for exercise is explained by the construct, suggesting that the scale items represent this construct well. Moreover, the SEE showed a strong association with levels of physical activity. This result aligns with the previous literature, indicating that those who feel more capable of exercising are more successful at overcoming barriers, thus increasing physical activity engagement [11,12,13,14]. Overall, these results support the convergent validity of the SEE.

Concerning nomological validity, the correlation matrix showed a significant association with the two indicators of physical and mental health and all five Big Five factors. According to the previous literature, older adults with better mental and physical health tend to feel more confident in exercising [22]. Our findings align with this hypothesis. Moreover, Judge and Ilies [55] found that extraversion, openness to experience, and conscientiousness were positively related to self-efficacy for exercise, while neuroticism was negatively related. Instead, agreeableness showed no significant relationship. In our study, we found a similar pattern, with the exception that the association between SEE and agreeableness was significant. These results could suggest a specific personality profile of older adults with high self-efficacy for exercise, similar to the one described by Picha and Howell [21].

Even if our findings provide evidence of the measurement invariance, reliability, and validity of the SEE, some limitations must be addressed. First, given the impact of cultural factors, additional studies are needed to generalize the results to other countries. Second, although our study found significant associations between self-efficacy for exercise and levels of physical activity and mental and physical health, a longitudinal research design is needed to further analyze the direction of these relationships.

## 5. Conclusions

Based on a sample of older adults from the U.S. and Italy, the present research provides evidence of the psychometric proprieties of a brief version of the SEE, suggesting that this questionnaire is a reliable and valid measure of self-efficacy for exercise in this population. The correlation of SEE with physical health (PCS), mental health (MCS), and the five scales of the BFI-2-XS linked this construct to mental and physical health, as well as to personality traits, supporting the nomological validity of the scale. Moreover, it suggests the possible existence of a specific profile for older adults with high self-efficacy for exercise.

Furthermore, our study established the full measurement invariance of the SEE across gender and age groups (i.e., 65–75 years old and over 75). This result suggests that, regardless of older people’s characteristics, they conceptualize and interpret the SEE items similarly. However, consistent with the previous literature suggesting the presence of cultural differences in barriers to physical activity [4,78,79], only partial scalar invariance of the SEE was achieved across countries, indicating the presence of minor cultural differences. Therefore, caution is needed in cross-cultural applications.

Given the crucial role of self-efficacy for exercise in promoting physical activity and well-being among older adults, evaluating the psychometric properties of an instrument aimed at measuring this perception represents a first important step in developing preventive interventions to foster exercise participation in this population. Understanding self-efficacy for exercise can also help identify individuals who may need additional support and motivation to engage in physical activity. Moreover, it allows healthcare providers to tailor interventions based on individual levels of confidence, monitor progress over time, and evaluate the effectiveness of programs aimed at enhancing physical activity and overall health. In this context, the SEE has been proven to be a reliable and valid measure for the assessment of self-efficacy for exercise in the older population. To further use the SEE in longitudinal studies, test-retest reliability and sensitivity to change need to be assessed.

## Figures and Tables

**Table 1 healthcare-12-01642-t001:** Mean, standard deviation (SD), skewness, and kurtosis of the SEE items.

	Item Content	Mean	SD	Skewness	Kurtosis
Item 1	Feeling pain when exercising	4.68	3.40	0.216	−1.221
Item 2	Did not enjoy the exercise	5.47	3.45	−0.076	−1.211
Item 3	Too busy for exercise	4.60	3.37	0.239	−1.095
Item 4	Feeling tired	4.62	3.38	0.231	−1.328
Item 5	Bad mood	5.60	5.60	−0.105	−1.328

Note: The SEE scores range from 0 to 10.

**Table 2 healthcare-12-01642-t002:** Goodness-of-fit indices for invariance of the SEE across older adults with different characteristics.

	S-B χ^2^	df	χ^2^/df	ΔS-B χ^2^	RMSEA	ΔRMSEA	CFI	ΔCFI	TLI	SRMR
U.S./Italian older adults										
Configural model	31.161	10	3.116		0.076		0.979		0.958	0.024
Metric model	45.931	14	3.281	16.612	0.079	0.003	0.969	−0.010	0.955	0.048
Scalar model	94.812	18	5.267	64.804	0.108	0.029	0.925	−0.044	0.916	0.059
Partial scalar model ^a^	69.282	17	4.075	29.818	0.092	0.013	0.949	−0.020	0.940	0.052
Partial scalar model ^b^	53.805	16	3.362	8.594	0.081	0.002	0.963	−0.006	0.954	0.046
Female/Male older adults										
Configural model	29.003	10	2.900		0.072		0.982		0.963	0.023
Metric model	37.379	14	2.670	5.923	0.068	−0.004	0.977	−0.005	0.968	0.032
Scalar model	47.913	18	2.662	10.480	0.068	0.000	0.971	−0.006	0.968	0.033
65–75 years old/over 75 older adults										
Configural model	34.004	10	3.400		0.081		0.977		0.955	0.026
Configural model ^c^	23.223	9	2.580		0.066		0.987		0.970	0.021
Metric model ^c^	31.308	13	2.408	6.819	0.062	−0.004	0.983	−0.004	0.973	0.031
Scalar model ^c^	37.801	17	2.224	5.477	0.058	−0.004	0.980	−0.003	0.977	0.035

Note: χ^2^, chi-squared; df, degree of freedom; χ^2^/df, normative chi-squared; RMSEA, root mean squared error of approximation; CFI, comparative fit index; TLI, Tucker–Lewis Index; SRMR, standardized root mean square residual. ^a^ Intercept of item 4 was freed. ^b^ Intercept of item 4 and item 3 were freed. ^c^ In the over 75 group, items 2 and 5 were free to covary. Loss of goodness-of-fit was tested by comparing the more restricted model with the less restricted model.

## Data Availability

Data are available upon request to the first author.

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
