# Peer review of "Measuring Self-Efficacy for Exercise among Older Adults: Psychometric Properties and Measurement Invariance of a Brief Version of the Self-Efficacy for Exercise (SEE) Scale"

_healthcare, 2024, doi:10.3390/healthcare12161642_

Round 1

Reviewer 1 Report

Comments and Suggestions for Authors

This manuscript presents the validation of a 5-item version of the Self-Efficacy for Exercise Scale (SEE) in individuals over 65 from the US and Italy. The one-factor model of the SEE demonstrated full scalar invariance across gender and age, and partial scalar invariance across cultures, with adequate reliability and expected relations to other variables. The theoretical background is clearly articulated, the data-analyses and results are well-presented, and the implications for promoting exercise participation among the elderly are discussed effectively.

I have only the following (minor) points:

Abstract

Information on reliability estimates should be included in the abstract

Introduction

Under section “1.3. SEE measurement invariance and cultural differences”, the relevance of testing for measurement invariance across cultures is well-explained. However, the rationale for exploring invariance across gender and age should also be clearly presented.

When explaining the levels of measurement invariance, the authors do not consider strict or residual invariance. This choice should be convincingly justified.

In “The present study section”, please provide a rationale for the chosen age groups (65-75 vs 76+).

In addition to the direction, please provide a hypothesis for the effect size of the correlation between the SEE and the GLSTPAQ. The same applies to the expected correlations with mental and physical health, conscientiousness, extraversion, and neuroticism.

Materials and methods

In the “Sample and procedure section”, please add a significance test for differences in gender and age across the US and ITA samples.

Please include more information on the setting and timing of recruitment in both the US and Italy. How were US and Italian participants approached? How were the exclusion criteria ascertained? Additionally, please also include information on participation rates.

In “Measures”, please provide the reliability estimates for each criterion measure.

Data analysis

At line 260, SLI should be LSI.

Please note that Online Resource 2 (mentioned at line 266) is not available on the susy platform.

Results

If the MLR estimation method was used for CFAs, please provide the S-B χ² (line 275 and Table 2).

Please include the range of values for the residuals in the main text (line 276).

Please add  the S-B Δχ² test for comparisons across nested models in both the Results and Data analysis sections.

A table note states that “cIn the over 75 group items 2 and 5 were free to covary”. Please add details on this in the main text, both in the Results and in the Discussion sections.

Section 3.3. Nomological validity: please comment on the effect size of the correlations.

Because (partial) scalar invariance was met, it would be worthwhile to compare the groups and test for the SEE known-group validity.

Discussion

As a direction for future research on the SEE’s psychometric functioning, it would be desirable to test its test-retest reliability and sensitivity to change.

Reviewer 2 Report

Comments and Suggestions for Authors

This is a solid paper that requires some minor modifications:

1. I question the categorization of demographics as a barrier. Being a given age or gender per se means little in light of motivational, physical or psychosocial barriers and has no relevance to intervention to enhance exercise. The lack of a partner with whom one might exercise is not discussed, especially relevant to older persons.

2. Avoid using the phrase "the elderly"-it is ageist in nature.

3. What is the rationale for choosing 75 as a cutting point? Likewise, why choose Italy? 

4. Implicitly, focusing on older adults invites a comparison with younger adults-why was this not done?

5. Acronyms (e.g. PA) should be avoided.

6. Given the cross-cultural nature of the comparisons, the authors need to do a better job of speaking to why they would expect a priori such differences in the psychometrics of the SEE scale to exist.

7. I think more clarity is needed in how the authors went from a 9-item scale to a 5-item scale.

Reviewer 3 Report

Comments and Suggestions for Authors

The study concerns the testing of psychometric characteristics of a Self-Efficacy Exercise [SEE] scale modified by the authors. The cross-cultural standardisation was conducted on many elderly subjects [two age groups] nearly equivalent in gender and belonging to two nationalities [USA and Italy]. The theoretical presentation of the study's relevance is extensive, articulate and well-documented. The instruments used and the methodology adopted are well explained and justified. The expectations of the researchers concerning the psychometric robustness of the new version of the SEE, the independence to age and gender and the importance of the cultural component in the self-assessment of one's relationship with physical activity [PA] are verified by the data analysed with appropriate statistical procedures.

Accurate description of results and well-documented discussion as well. There is a problem with the editing of Table 2 in which the column labels are not aligned with the relevant data. Furthermore, in the note at the foot of the table, the explanatory information relating to notes a, b, and c does not appear separated: in particular, the letters identifying specific results in the table above are attached to the relevant explanatory sentence. All this makes reading less immediate.
